# Spatio-Temporal Changes in Wildlife Habitat Quality in the Greater Serengeti Ecosystem

**Hamza K. Kija [1,2,\*], Joseph O. Ogutu [3], Lazaro J. Mangewa [1], John Bukombe [2], Francesca Verones [4], Bente J. Graae [5], Jafari R. Kideghesho [6], Mohammed Y. Said [5,7] and Emmanuel F. Nzunda [8]**

[1] Department of Wildlife Management, College of Forestry, Wildlife and Tourism, Sokoine University of Agriculture (SUA), P.O. Box 3073, Morogoro, Tanzania; ljohannah@yahoo.co.uk

[2] Tanzania Wildlife Research Institute (TAWIRI), Conservation Information and Monitoring Unit (CIMU), P.O. Box 661, Arusha, Tanzania; bukombe2017@gmail.com

[3] Biostatistics Unit, Institute of Crop Science, University of Hohenheim, Fruwirthstr. 23, 70599 Stuttgart, Germany; jogutu2007@gmail.com

[4] Department of Energy and Process Engineering, Norwegian University of Science and Technology, N-7491 Trondheim, Norway; francesca.verones@ntnu.no

[5] Department of Biology, Norwegian University of Science and Technology, N-7491 Trondheim, Norway; bente.j.graae@ntnu.no (B.J.G.); msaid362@gmail.com (M.Y.S.)

[6] College of African Wildlife Management (CAWM), P.O. Box 3031, Moshi, Tanzania; kideghesho@yahoo.com

[7] Institute for Climate Change and Adaptation, University of Nairobi, P.O. Box 30197, Nairobi 00100, Kenya

[8] Department of Forest Resources Assessment and Management, College of Forestry, Wildlife and Tourism, Sokoine University of Agriculture (SUA), P.O. Box 3013, Morogoro, Tanzania; nzunda@sua.ac.tz

\* Correspondence: hamza.kija@tawiri.or.tz; Tel.: +255-768-611844

**Abstract:** Understanding habitat quality and its dynamics is imperative for maintaining healthy wildlife populations and ecosystems. We mapped and evaluated changes in habitat quality (1975–2015) in the Greater Serengeti Ecosystem of northern Tanzania using the Integrated Valuation of Environmental Services and Tradeoffs (InVEST) model. This is the first habitat quality assessment of its kind for this ecosystem. We characterized changes in habitat quality in the ecosystem and in a 30 kilometer buffer area. Four habitat quality classes (poor, low, medium and high) were identified and their coverage quantified. Overall (1975–2015), habitat quality declined over time but at rates that were higher for habitats with lower protection level or lower initial quality. As a result, habitat quality deteriorated the most in the unprotected and human-dominated buffer area surrounding the ecosystem, at intermediate rates in the less heavily protected Wildlife Management Areas, Game Controlled Areas, Game Reserves and the Ngorongoro Conservation Area and the least in the most heavily protected Serengeti National Park. The deterioration in habitat quality over time was attributed primarily to anthropogenic activities and major land use policy changes. Effective implementation of land use plans, robust and far-sighted institutional arrangements, adaptive legal and policy instruments are essential to sustaining high habitat quality in contexts of rapid human population growth.

**Keywords:** Serengeti ecosystem; threats; InVEST model; protected areas; savannah; quality; buffer

## 1. Introduction

Quality wildlife habitats, the areas that provide shelter and forage and support survival of wildlife species, are declining worldwide [1]. Habitat quality is the ability of the land to provide essential habitat components for a particular species, and is among the key ecological attributes that determine

wildlife population status in landscape [2]. The quality of a habitat influences wildlife species diversity, density, distribution, and movement patterns in landscapes [2,3]. Habitat quality is influenced by multiple factors, mainly human-induced pressures, including over-utilization of biodiversity resources, poor land use practices, and climate change. Overexploitation of biodiversity stems from human population growth and the associated socio-economic development activities, such as expansion of settlements and agriculture within or close to wildlife habitats. This typically results in wildlife habitat degradation, including deforestation [4], changes in land use and land cover [5], fragmentation [6], and, ultimately, deterioration of habitat quality, leading to biodiversity loss [7]. Since high habitat quality is a key determinant of vibrant wildlife populations in any ecosystem [8], its decline reduces the ability of habitats to sustain diverse wildlife resources, resulting in altered species distributions, composition, and population abundance [9,10], and ultimately in loss of wildlife [11,12].

Decline in habitat quality is mainly attributed to inadequate management of the complex interactions between humans and wildlife habitats [13]. For instance, the 21st century has witnessed rapid expansion of agricultural production in the developed world to enhance food security for the increasing human population [14]. Human population growth is accelerating habitat fragmentation through land use and cover changes in temperate [15,16], tropical [5], and subtropical [17,18] regions. This constitutes the principal threat to most species. Furthermore, in the temperate zones, human-induced habitat loss following habitat fragmentation due to roads and related developments is considered the principal threat to most species [19,20]. Similar effects on wildlife habitats have been caused by agricultural expansion in the United States of America, thus degrading wildlife habitats [21]. Also, human and naturally mediated climate change is yet another important driver of wildlife habitat loss [22]. The interactions between biotic and abiotic factors, such as land use and climate change, also affect wildlife habitats at various spatio-temporal scales [23].

Habitat quality assessments provide scientific information on changes to environments as well as their drivers as a basis for making improvements to wildlife habitats [24]. Changes in habitat quality have serious implications for wildlife conservation, especially in savannah ecosystems. Savannah ecosystems, including in East Africa, are key habitats for diverse wildlife species and home to many iconic parks and wildlife migrations [25,26] but are experiencing habitat loss [27]. Unregulated conversions of forests (including through logging) and rangelands for crop production, mining, urban, and infrastructure development, among other human-induced changes, have led to declines in high quality wildlife habitats worldwide. These are often manifested by habitat loss, degradation of catchments and soil erosion, blocked migratory corridors and dispersal areas [28], low species diversity [29], and loss of wildlife species [30], leading to wildlife species extinctions, loss of biodiversity and human livelihoods [31].

In East Africa, the Serengeti ecosystem in Northern Tanzania is experiencing significant land-use changes, particularly due to increased anthropogenic pressures along its borders, mainly agriculture and settlements, and changes in various national policies [27]. The changes in land use and cover can adversely alter wildlife habitat quality and availability, and constrain ecological processes [32], or decrease the size of protected areas, diminish their ecological integrity and values, thus reducing ecosystem resilience [33,34]. Despite the declining habitat quality in the ecosystem and its surrounding buffer area, no study has quantified spatial and temporal changes in habitat quality in relation to the level of protection (Conservation Area, National Park, Game Reserves, Game Controlled Areas and Wildlife Management Areas) and the buffer area (30 km). Consequently, no previous study has examined the direction and magnitude of changes in habitat quality over time for various habitat types under contrasting levels of protection in the ecosystem. Few previous studies have examined changes in specific habitat components or habitat quality for specific wildlife species. For instance, [35] analyzed changes in lion (*Panthera leo*) population density and fitness as indicators of changes in their habitat quality in the Serengeti ecosystem and showed that density alone cannot sufficiently characterize changes in habitat quality for lions. Driving factors for ecological impacts of habitat destruction were

reviewed in the Serengeti ecosystem [8] and a previous study quantified changes in land cover and use and human population growth in the ecosystem during 1984–2003 [5].

The aim of this study was to map and quantify changes in wildlife habitat quality in space and time in the Greater Serengeti Ecosystem (GSE) using the Integrated Valuation of Environmental Services and Tradeoffs (InVEST) model. Understanding spatial and temporal dynamics of habitat quality is fundamental to effective management of ecosystems experiencing changing land use and cover. More precisely, we evaluated changes in habitat quality at three points in time (1975, 1995, and 2015) to provide a quantitative basis for improving wildlife habitat management and enhancing natural ecological processes in the GSE and possibly elsewhere.

## 2. Materials and Methods

### 2.1. Study Area

The Greater Serengeti Ecosystem (GSE) is located in northern Tanzania, within the Greater Serengeti-Mara Ecosystem (GMSE) that is defined by movements of migratory wildebeest (*Connochaetes taurinus*) and the common zebra (*Equus quagga buchellii*) between the Serengeti National Park in Tanzania and the Maasai Mara National Reserve in Kenya. The Serengeti ecosystem (1°24′57.16′′ S to 3°45′35.31′′ S and 33°52′0.15′′ E to 35°59′41.53′′ E) covers 33,106 km$^2$. It is world-famous for its wildebeest, zebra and Thomson's gazelle (*Eudorcas thomsonii*) migration, the largest terrestrial migration of large herbivores remaining on Earth [36–38]. This migration involves about 1.3 million wildebeest, 0.2 million zebra and 0.4 million Thomson's gazelle [39]. The migration is mostly confined within protected areas, notably the Serengeti National Park, Ngorongoro Conservation Area, Maswa, Grumeti and Ikorongo Game Reserves and Ikoma Wildlife Management Area in Tanzania and the Masai Mara National Reserve in Kenya (Figure 1).

The ecosystem has diverse vegetation cover, land cover, uses, and management types, including protected areas, traditional pastoral, agro-pastoral or cultivated areas (large and small-scale farms), forests, and wildlife conservation areas [39]. Anthropogenic activities strongly influence ecological processes and ecosystem structure within the protected areas. As a result, land use and cover changes in the buffer area exert substantial impacts on the flora and fauna inside the protected areas [23]. We defined a 30 km buffer (23,487 km$^2$) around the protected area to analyze edge effects linked to surrounding anthropogenic activities. The whole study area therefore covers about 56,593 km$^2$ (Figure 1).

Broadly, the study area has nine administrative units with different management regimes, ranging from total exclusion of human activities in National Parks and Game Reserves (IUCN category II), multiple land use with nationally controlled human activities, such as in the Ngorongoro Conservation Area where livestock grazing and human settlement are allowed (IUCN category V and VI) and locally controlled use of resources (Wildlife Management Areas) and free use zones (the 30 km buffer area) in which livestock grazing, agriculture, human settlements, and other land uses are not restricted (Figure 1). The Ngorongoro Conservation Area was declared a UNESCO Man and Biosphere Reserve in 1971 and a World Heritage Site in 1979 whereas the Serengeti National Park was declared a UNESCO World Heritage Site in 1981.

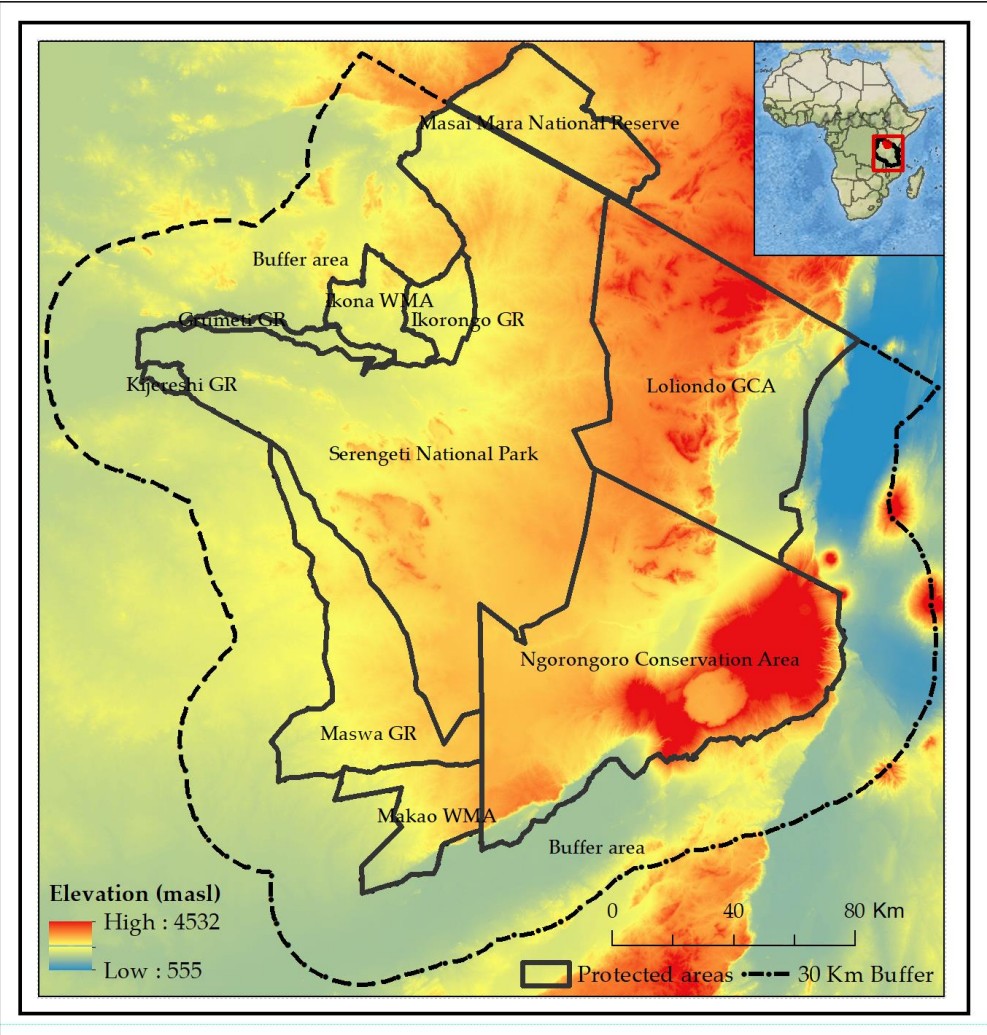

**Figure 1.** Map of the study area (the red dot in the small inset shows the location of the study area in East Africa). The red box shows the location of Tanzania and southern Kenya in Africa.

## 2.2. Modelling Habitat Quality Using the InVEST Model

We used the Integrated Valuation of Environmental Services and Tradeoffs (InVEST) model to evaluate changes in habitat quality in the ecosystem. InVEST is a spatially-explicit geographic information system (GIS) based tool for evaluating habitat condition in terrestrial or aquatic ecosystems based on data from land use and land cover (LULC) maps and biophysical factors. The InVEST model assumes that high-quality habitat supports high biodiversity as opposed to areas with low or poor-quality habitats [40–42]. The InVEST model evaluates habitat condition based on (i) relative impacts of each threat, (ii) habitat sensitivity to the threat, (iii) the distance between the threat and the habitat, and (iv) the degree to which the land is legally protected [43].

## 2.3. Data Requirements and Sources

The InVEST model uses maps as information sources and produces maps as outputs. Table 1 shows the model's data requirements. These are described in detail in Section 2.3.1 to Section 2.3.5 below.

**Table 1.** Input data for the Integrated Valuation of Environmental Services and Tradeoffs (InVEST) habitat quality model.

| Data Type | Data Description |
|---|---|
| 1.　Land use and cover map (LULC) | Land use and cover map for the study ecosystem. |
| 2.　Threat data | Raster data with the distribution and intensity of each habitat threat. |
| 3.　Legal accessibility | Data showing the legally recognized level of protection for each habitat. |
| 4.　The sensitivity of habitat types to each habitat threat | LULC table with habitat types and their sensitivities on a scale from 0–1, with 1 denoting the highest possible level of habitat sensitivity [44,45]. |
| 5.　Half saturation constant | Determined half-saturation value at 0.5 [4]. |

### 2.3.1. Land Use and Land Cover (LULC) Data

Landsat Multispectral Scanner (MSS), Thematic Mapper (TM) and Landsat-8 (Operational Land Imagery) time series images for the years 1975, 1995 and 2015 corresponding to paths 168, 169 and 170 and rows 61, 62 and 63 of the Landsat Worldwide Reference System (WRS) were downloaded from the Earth Explorer (https://earthexplorer.usgs.gov/) web platform. The potential effects on the images of seasonal differences in vegetation phenology between the three periods were minimized by downloading image scenes captured on similar satellite overpass times or seasons, particularly dry periods spanning January–February, July–August and October–November. Dry season images are preferable to wet season images because they have relatively lower cloud cover.

Approximately 1918 training and validation samples were collected at randomly selected sampling points in the GSE between 2015 and 2016. For historical images, the Herlocker (1976) and Reed et al. [46] were used to obtain the training and testing sets for the 1975 and 1995 imageries, respectively. The images were pre-processed and classified using the collected samples through the random forest (RF) classification algorithm [47]. Eight land cover classes, including woodland, shrubland, grassland, wetland and swamps, water bodies, settlement, agriculture, and bareland, were distinguished. The classifier achieved accuracies of 88.4%, 90.6%, and 93.4% with Kappa Indices of Agreement of 0.86, 0.87, and 0.91 for 1975, 1995, and 2015 thematic maps, respectively, which are adequate for most practical applications. Most of the natural vegetation constitutes key wildlife habitats [48]. Furthermore, the mapped LULC are known to support a wide array of wildlife species. Woodland and mosaic (shrubland, grassland and riverine) habitats were considered key habitats for various wildlife species [49], and thus were assigned the highest suitability index. For aquatic habitats, such as wetlands and swamps, suitability increases with increasing size. Anthropogenic land cover types such as bareland, settlements and agricultural areas, roads, and other features were considered unsuitable for wildlife. We assigned either 0 or 1 (binary index) to each LULC class, with 0 indicating unsuitable and 1 denoting highly suitable habitats.

### 2.3.2. Habitat Threat Data

We identified and selected seven anthropogenic threats to habitats (Table 2) in the protected area and human-dominated buffer in the study ecosystem. The selected four threat factors are roads, rivers, major urban centers [2,50], and cultivated plus built-up areas. Roads (paved and unpaved) can threaten habitat integrity by restricting wildlife movements between habitat patches [51–53], enhancing accessibility of protected areas (e.g., for poachers), providing pathways for invasive species [54], and accelerating habitat loss and fragmentation and hence biodiversity loss [55]. Highly utilized rivers and other water sources can threaten habitats in the ecosystem by attracting human activities, which, in turn, depress habitat quality and threaten wildlife species [50] and rank among the leading causes of contemporary global biodiversity loss [3,56]. The habitat threat data were obtained from various sources. Data on roads and rivers for 1975, 1995, and 2015 were obtained from the Tanzania National Bureau of Statistics (TNBS) database (https://www.nbs.go.tz/), DIVA-GIS (http://www.diva-gis.org/), Africover, AfricanBioServices project (https://africanbioservices.eu/), digitization of standard topo sheets (1:50,000), especially for 1975, and fieldwork (2016–2018). Major urban centers, cultivated and

built-up areas were also considered as habitat threats as they degrade natural habitats [57]. Data on cultivated and built-up areas were extracted from LULC maps for 1975, 1995, and 2015.

**Table 2.** Ecological habitat threat factors.

| Threat Factor | Maximum Distance (Km) | Weight | Decay |
|---|---|---|---|
| Population | 25 | 1 | Exponential |
| Agriculture | 15 | 0.8 | Exponential |
| Paved road | 20 | 0.5 | Linear |
| Unpaved road | 27 | 0.6 | Linear |
| Permanent river | 15 | 0.5 | Exponential |
| Seasonal river | 10 | 0.3 | Exponential |
| Livestock distribution | 8 | 0.4 | Exponential |

Note: The maximum distances are based on values modified from [5,52,53,58–60].

The threat factors summarized in Table 2 show habitat-specific threats, impact distance (at which each threat would affect the habitat) and scale (ranging from 0 to 1), which quantifies the impact of each threat factor on habitat quality relative to other threats in our strata which comprises wildlife management areas, game controlled areas, game reserves, conservation areas, and national parks.

### 2.3.3. Legal Accessibility

Legal accessibility is the degree to which land is legally protected against anthropogenic activities. Legal accessibility is important in assessing habitat quality of a landscape [59]. The InVEST model assumes that the legal accessibility of land is important in minimizing threats as it is typically accompanied with administrative policies or management plans and that protected areas are less impacted than non-legally protected areas [44]. We obtained spatial data on protected areas from the Tanzania Wildlife Research Institute (TAWIRI) and TNBS [27]. Based on the status of their protection, we determined the value of protection status as indicated in Table 3. Wildlife management areas (WMAs) were excluded from the 1975 and 1995 analyses because they came into existence in 2007. Kijereshi was a WMA in 1994 before it was upgraded to a game reserve in 1996.

**Table 3.** Weights assigned to legal accessibility.

| Category | Weight |
|---|---|
| Buffer | 1 |
| Game Controlled Area | 0.50 |
| Wildlife Management Area | 0.40 |
| Conservation Area | 0.35 |
| Game Reserve | 0.25 |
| National Park | 0.15 |

### 2.3.4. Habitat Sensitivity

Each habitat type is assumed to be sensitive to a particular threat and not all habitat threats have equal impacts on all habitat types. We converted the threats data (Section 2.3.2) from a vector to a raster data format using ArcGIS 10.5 (ESRI 2019). We normalized and ranked threat maps on a threat intensity scale ranging from 0 to 1, where 0 represents the least and 1 the highest threat level. For example, a given habitat or LULC would score 0 if it is not and 1 if it is sensitive to a specific threat factor, respectively [44].

Information on how habitats in the study area respond to threat factors was either unavailable or only partially available. Therefore, we sought expert knowledge of likely habitat responses to the threat factors from ecologists who had worked in the ecosystem. Based on this, we generated a sensitivity table (Table 4) showing the sensitivity of each LULC class to specific habitat (Section 2.3.1) threat factors. The impact of a threat factor on a given habitat was determined by several factors,

including the distance from the habitat type to the threat; the closer a threat is to a habitat, the higher its expected impact [2,44].

**Table 4.** Habitat suitability and sensitivity of each land-use type to ecological threat factors.

| Land Use Type | Habitat | Sensitivity to | | | | | | |
|---|---|---|---|---|---|---|---|---|
| | | Agric | Pvdrd | Unpvrd | Prvs | Srvs | Popn | Lvst |
| Bareland | 0 | 0 | 0 | 0 | 0 | 0 | 0 | 0 |
| Agriculture | 0 | 0 | 0.1 | 0.2 | 0.7 | 0.4 | 0.5 | 0.2 |
| Settlement | 0 | 0.8 | 0.2 | 0.5 | 0.6 | 0.3 | 0.6 | 0.3 |
| Water bodies | 1 | 0.5 | 0.2 | 0.2 | 0 | 0 | 0.8 | 0.2 |
| Grassland | 1 | 0.9 | 0.5 | 0.7 | 0.6 | 0.4 | 0.7 | 0.8 |
| Shrubland | 0.6 | 0.5 | 0.2 | 0.3 | 0.3 | 0.2 | 0.6 | 0.3 |
| Woodland | 0.8 | 0.8 | 0.4 | 0.6 | 0.5 | 0.3 | 0.7 | 0.2 |
| Wetland and Swamps | 0.6 | 0.6 | 0.5 | 0.5 | 0.4 | 0.2 | 0.7 | 0.6 |

Agric = agriculture, Pvdrd = paved road, Unpvrd = unpaved road, Prvs = permanent river, Srvs = seasonal river, Popn = population, and Lvst = livestock.

We resampled the LULC and other data layers to 500 m × 500 m resolution. We chose this resolution based on compound analysis of the study area size and computational feasibility. The study area size of 56,600 km$^2$ yielded approximately 5.05 billion cells at a 30 m × 30 m resolution, which was computationally too expensive to process. We reduced computational complexity by resampling the pixels to 2.02 billion cells at a 500 m × 500 m resolution for further analyses.

The resampled datasets were projected to the Universe Transverse Mercator (UTM) Zone 36 South using the World Geodetic System (WGS) 1984. In addition, all raster datasets were re-sampled to a common 500 m grid for ease of processing.

### 2.3.5. Half-Saturation Constant

Half or semi-saturation constant $k$ is used by the InVEST model to convert the habitat degradation score into habitat quality score. The value of $k$ is normally set equal to half (0.5) of the grid cell size [42,48].

### 2.4. Data Analysis

Modelling Habitat Quality Using the InVEST Model

Habitat quality is the ability of the environment to provide essential conditions required for sustenance or persistence of an individual organism [61] and hence is a major determinant of landscape biodiversity. Larger habitats support more biodiversity and vice-versa. Furthermore, the more degraded an area is, the lower is the biodiversity value it can hold [2]. Habitat quality is assumed to depend upon the relative impact of threats, the sensitivity of habitats to the threats, the distance between a habitat and sources of the threats, and location of protected areas. The model uses an exponential decay function to describe the impact $i_{rxy}$ of threat $r$ from a grid cell $y$ on a habitat in grid cell $x$, located at a linear distance $d_{xy}$ from a threat source [48]:

$$i_{rxy} = \exp\left[-\left(\frac{2.99}{d_{rmax}}\right)d_{xy}\right] \tag{1}$$

The total threat level $D_{xj}$ in a grid cell $x$ with LULC $j$ is then calculated as:

$$D_{xj} = \sum_{1}^{R}\sum_{1}^{Y}\left(\frac{W_r}{\sum_{1}^{R} W_r}\right)r_y i_{rxy}\beta_x S_{jr} \tag{2}$$

where $R$ is the number of all the ecological threat factors, $D_{xj}$ is the set of all grid cells on raster map $r$, $W_r$ is the threat weight that relates the destructiveness of a degradation source to all habitats, $r_y$ indexes all grid cells in raster map $r$, $i_{rxy}$ is a function describing the exponential decay in habitat quality as a function of distance from ecological threat factors, $\beta_x$ is the level of accessibility to grid cell $x$, with 1 indicating complete accessibility and 0 complete inaccessibility, and $S_{jr}$ is the sensitivity of land use type $j$ (habitat type) to the ecological threat factor $r$.

Finally, the habitat quality $Q_{xj}$ of LULC $j$ is calculated based on the habitat suitability of LULC $j$ using [48,62]:

$$Q_{xj} = H_j \times \left[ 1 - \left( \frac{D_{xj}^z}{D_{xj}^z + k^z} \right) \right] \tag{3}$$

where $H_j$ is a habitat quality score ranging from 0 to 1, with non-habitat land use and cover types given a score of 0 and perfect habitat classes a score of 1, $D_{xj}$ is the total threat level in grid cell $x$ with land use and cover type $j$, $k$ is the half-saturation constant, and $z$ is a constant.

Using the natural breaks (Jenks) classification method, we sliced the resultant habitat quality score map into four habitat class categories: poor, low, medium, and high. Natural breaks is a standard classification method that arranges dataset values into different internally homogenous classes [63].

## 3. Results

### 3.1. Habitat Quality (1975–2015)

The ecosystem was dominated by low-quality followed by high-quality habitats in 1975 (Figures 2 and 3). In the subsequent 21 years (1975–1995), the poor-quality habitats increased four-fold, the low-quality habitats reduced by a half, and the medium-quality habitats doubled in size, but the high-quality habitats hardly changed relative to 1975 (Figures 2 and 3). In the 21 years leading to 2015 (1995–2015) the poor-quality habitats doubled, the low-quality habitats reduced to 0.59 times, and the medium-and high-quality habitats reduced to 0.62 times, whereas the high-quality habitat increased 1.3 times relative to 1995 (Figures 2 and 3). Overall, most of the poor-quality habitats in 1975 became either poor- or medium-quality habitats during 1975–1995 but most low-quality habitats in 1995 were reduced to poor-quality habitats during 1995–2015. Over the entire study period (1975–2015), a large part of the low-quality habitats was degraded to poor-quality habitats in the ecosystem and its surrounding buffer zone (Figure 3).

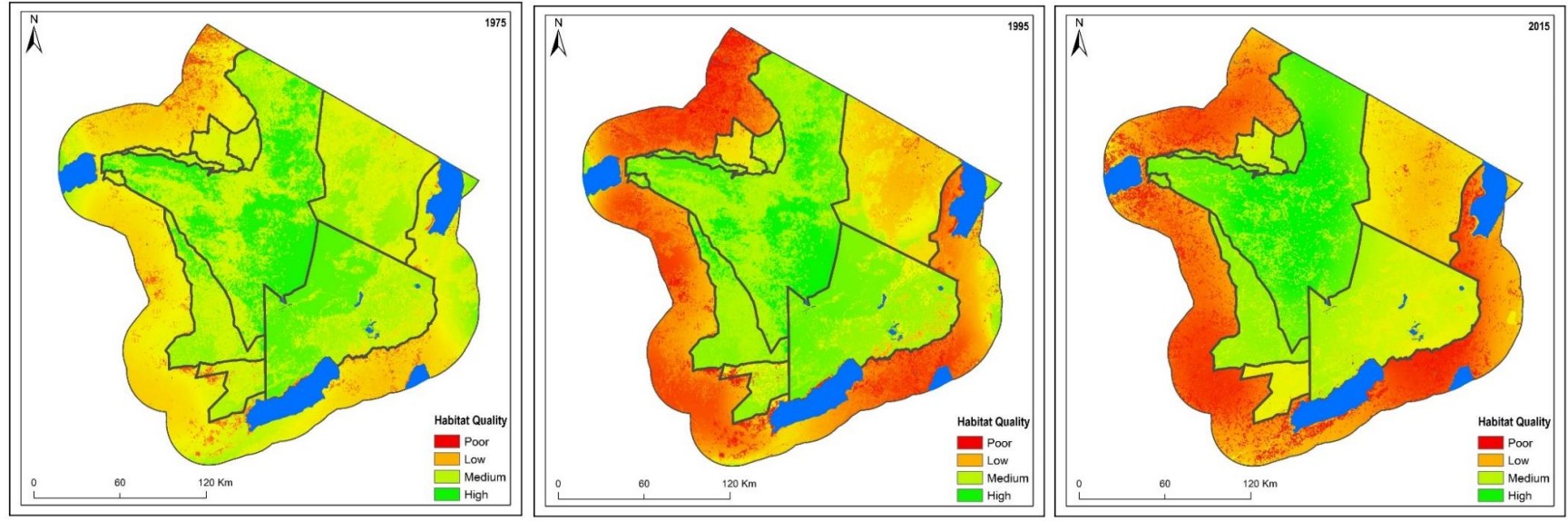

**Figure 2.** The spatial distribution of the four habitat quality classes in the ecosystem and the surrounding buffer zone in 1975, 1995, and 2015.

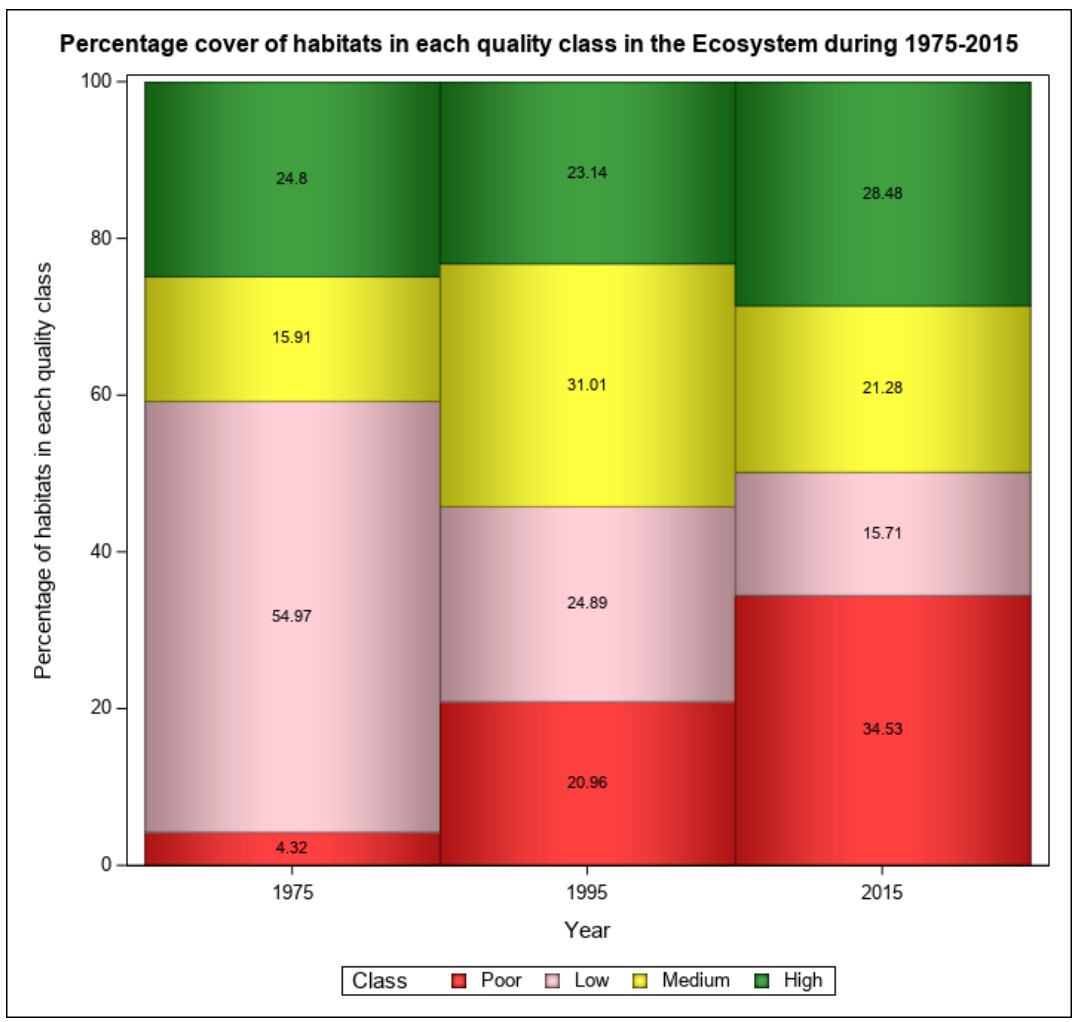

**Figure 3.** Cumulative percentage cover of habitats in each of the four quality classes in the ecosystem and surrounding 30 km buffer zone in 1975, 1995, and 2015.

At the stratum level in 1975, the high-quality habitats were the most prevalent in the WMAs and the national park (NP), which also had large coverage of the low-quality habitats, while low-quality habitats dominated the buffer zone (BF), conservation area (CA) and the game reserves (GR) in 1975 (Figure 4). In 1995, poor- and low-quality habitats characterized the buffer zone, medium-quality habitats dominated the GCA, medium- and high-quality habitats dominated the CA, WMAs and the national park, and low- and medium-quality habitats predominated in the GRs (Figure 4). In 2015, poor-quality habitats dominated the buffer zone, medium-quality habitats dominated the GCA and WMAs, high-quality habitats dominated the CA and the national park, and low-quality habitats dominated the GRs (Figure 4). Across 1975–2015, habitat quality declined most markedly in the buffer zone. Generally, areas with greater protection had relatively large proportions of medium and high-quality habitats.

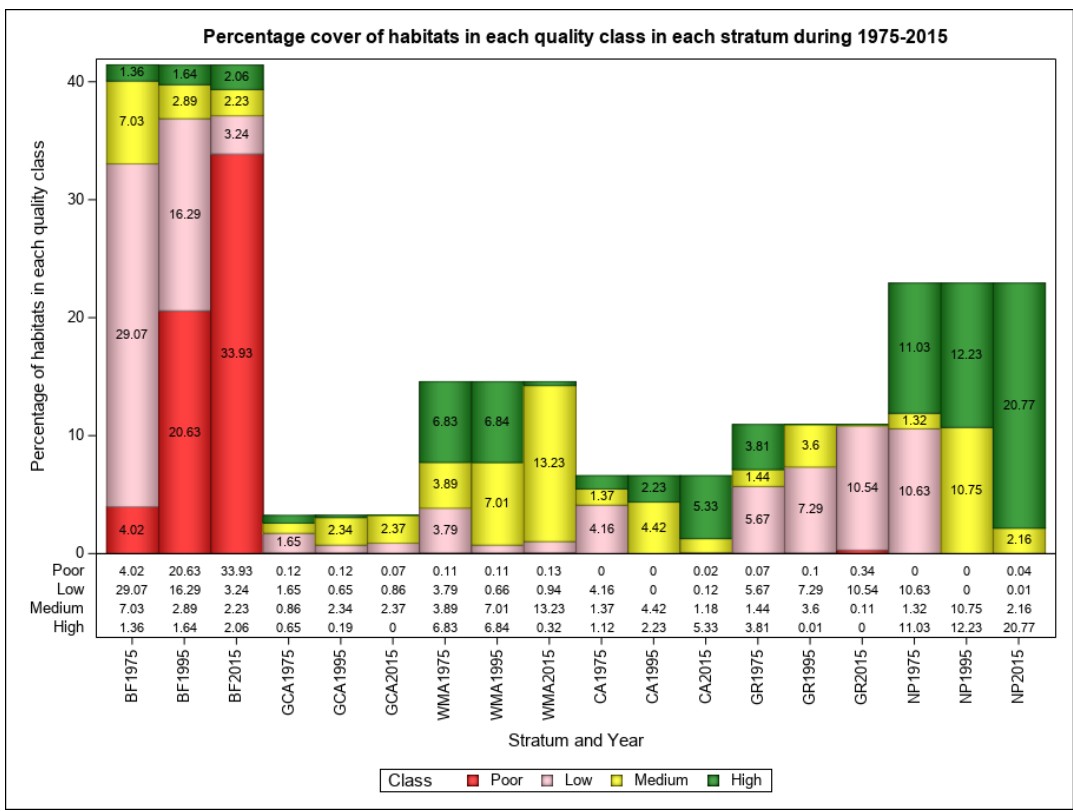

**Figure 4.** Cumulative percentage cover of habitats in each of the four quality classes for each stratum in the ecosystem and surrounding 30 km buffer zone in 1975, 1995, and 2015. BF = buffer zone, GCA = game controlled area, WMA = wildlife management areas, CA = conservation area, GR = game reserves, NP = national park.

### 3.2. Changes in Habitat Quality During 1975–1995, 1995–2015, 1975–2015

At the level of the ecosystem and its surrounding buffer area, poor- and medium-quality habitats increased whereas low- and high-quality habitats decreased during 1975–1995. During 1995–2015, both poor- and high-quality habitats increased whereas low- and medium-quality habitats declined. Across the 41-year study period (1975–2015), poor-quality habitats increased 2.5-fold, medium- and high-quality habitats increased slightly, and low-quality habitats declined markedly in the ecosystem and its adjacent buffer area (Figure 5). Most of the decline in low-quality habitats during 1975–2015 occurred during 1975–1995 (Figure 5).

Across the individual strata, poor-quality habitats increased most remarkably in the buffer area throughout 1975–2015 (Figure 6). In the buffer area, poor-quality habitats increased remarkably during 1975–2015 but declined slightly in 2015 relative to 1995. Low- and medium-quality habitats declined in the buffer area throughout 1975–2015 (Figure 6). In the GCA, only minor changes occurred in habitat quality during 1975–2015. In the WMAs, medium-quality habitats expanded whereas low- and high-quality habitats contracted throughout 1975–2015 (Figure 6).

In the CA, high-quality habitats increased four-fold whereas low- and medium-quality habitats declined during 1975–2015. In the GRs, low-quality habitats increased whereas high-quality habitats declined during 1975–1995. Poor- and low-quality habitats expanded whereas medium- and high-quality habitats reduced during 1975–2015 (Figure 6).

In the Serengeti National Park, the high-quality habitats increased two-fold whereas low- and medium-quality habitats declined markedly during 1975–2015 (Figure 6).

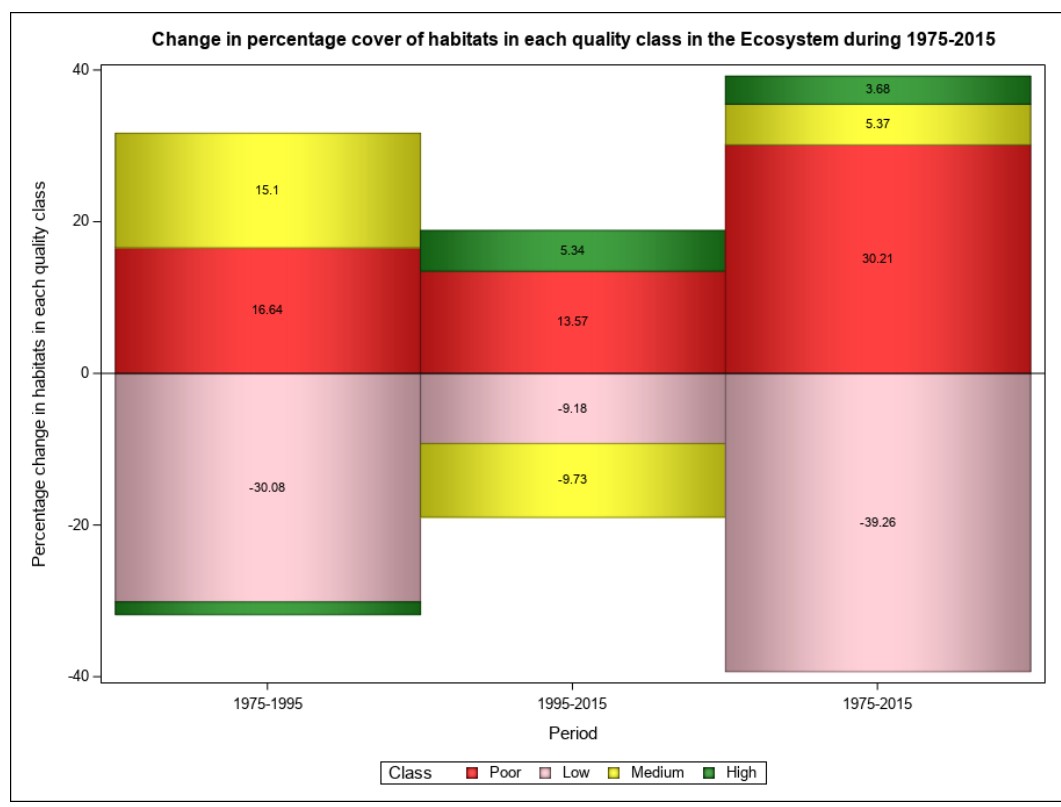

**Figure 5.** Changes in percentage cover of habitats in each of the four quality classes in the entire ecosystem and surrounding 30 km buffer zone during 1975–1995, 1995–2015, and 1975–2015. BF = buffer zone, GCA = game controlled area, WMA = wildlife management areas, CA = conservation area, GR = game reserves, NP = national park.

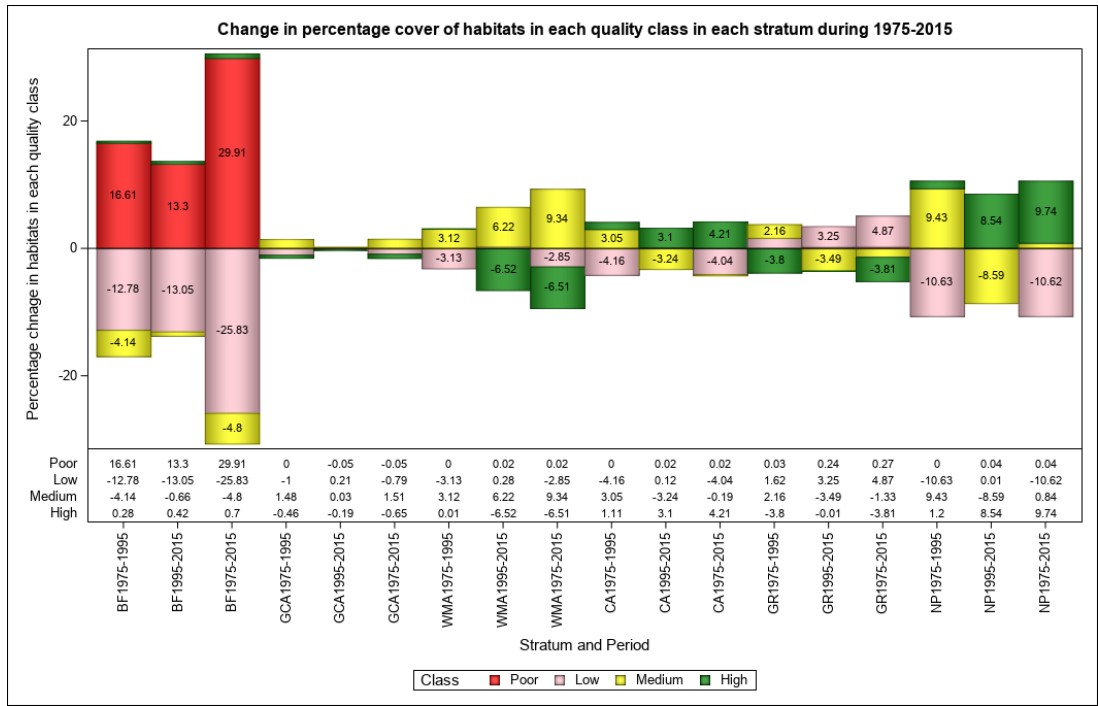

**Figure 6.** Changes in percentage cover of habitats in each of the four quality classes for each stratum in the ecosystem and surrounding 30 km buffer zone during 1975–1995, 1995–2015, and 1975–2015.

### 3.3. Transformations of Habitat Quality Classes (1975–1995, 1995–2015 and 1975–2015)

During 1975–1995 the ecosystem and its buffer area experienced high transformation of the low-quality to poor- and medium-quality habitats (Figure 7), whereas the medium-quality habitats were partially transformed to low-quality habitats. The high-quality habitats were partly transformed to the medium class. However, during 1995–2015, more low-quality habitats were transformed to the poor-quality class, and much of the medium-quality to high-quality habitats (Figure 7). Overall (1975–2015), most of the low-quality habitats were transformed to the other quality classes, predominantly the poor-quality class (Figure 7). Generally, the high-quality habitats decreased slightly because of contributions to the other classes.

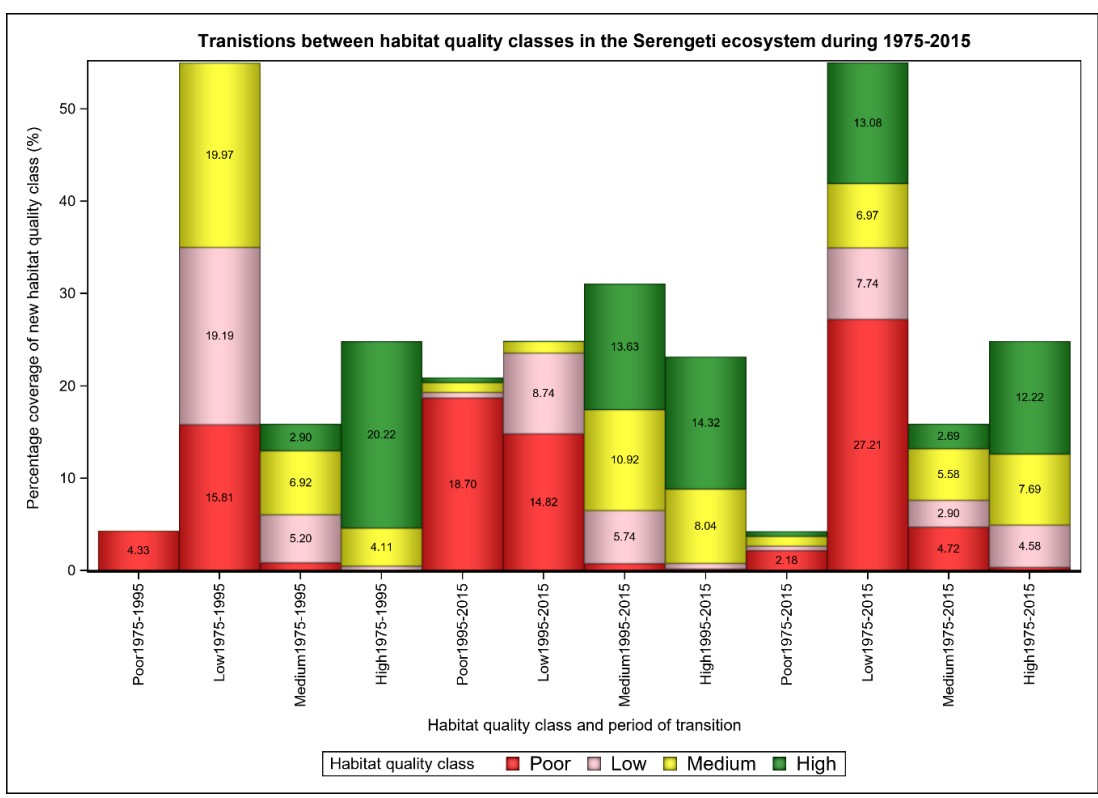

**Figure 7.** Transitions in habitat quality classes in the Greater Serengeti Ecosystem (GSE) during 1975–2015.

## 4. Discussion

We found substantial changes in the overall habitat quality in the GSE and its surrounding buffer area during the 41-year study period (1975–2015). This was manifested in an overall eight-fold increase in poor-quality habitats but only 1.2-fold increase in high-quality habitats. Moreover, low-quality habitats declined three-fold whereas the medium-quality habitats first increased two-fold and then declined by a quarter. The greatest overall decline in habitat quality occurred in the buffer area followed by the protected areas with low protection status (GCA, WMA, GRs). The Ngorongoro Conservation Area (CA) with a medium conservation status had an overall three-fold increase in the medium quality habitats whereas the most strictly protected Serengeti National Park had an overall eight-fold increase in the high-quality habitats during the 41-year study period (1975–2015). We summarize the detailed patterns of changes in habitat quality for the GSE and its surrounding buffer area (30 km) next.

### Habitat Quality (1975–2015)

The high proportion of low-quality habitats in 1975 can mainly be attributed to the Arusha Declaration of 1967 and Iringa Declaration (*Siasa ni Kilimo*, i.e., Politics is Agriculture) of 1972 both of

which promoted agricultural production as the economic backbone of Tanzania [64]. Furthermore, the villagization policy initiated in 1974 encouraged local communities to move and establish new settlements in collective villages, called "Ujamaa Villages", to enhance social services and common use of tools of agricultural production, such as tractors [65–67].

The policy was implemented through a large-scale resettlement program across mainland Tanzania, which had significant environmental implications [65,66]. More lands were opened up for establishment of new farms for various cash crops (cotton, sisal, and coffee), and food crops (paddy rice, wheat, and maize) depending on the local agro-climatic conditions. In consequence, the villagization policy led to significant changes in land use, soil erosion and land degradation in the country [66]. In 2012, changes in land use and land cover around the GSE were probably responsible for changes in habitat quality [5].

The improvements in habitat quality through 1975–1995 can be attributed to various conservation and environmental management measures. For example, farmers were encouraged by the Government of Tanzania to adopt and practice modern agricultural techniques with low environmental impacts [65]. Nevertheless, low agricultural production due largely to challenges associated with implementing the policy, including distance to the initial farms and climatic conditions, resulted in only slight improvements to habitats quality [68–70]. Besides, areas such as Kijereshi, Ikorongo, and Grumeti that were initially open areas (OAs) were upgraded to game controlled areas (GCA) [27]. Despite the relatively high proportion of low-quality habitats by 2015, various conservation efforts plus strengthened legal and institutional frameworks contributed to increased coverage of the medium- and high-quality habitats. These included upgrading of the GCAs to game reserves and establishment of wildlife management areas (Ikona and Makao WMAs). By 1975, the only protected areas in the ecosystem were the Serengeti NP, the Maswa GR, and the Loliondo GCA as the other areas were under a non-protected status.

Between 1975 and 1995, three game reserves (Kijereshi, Ikorongo, and Grumeti) were established leading to restricted access to livestock grazing and agriculture [27]. This is consistent with the increase in habitat quality with upgrading of the conservation status of protected areas in the ecosystem. As a result, buffer areas with the weakest protection status, followed by GCA, were dominated by poor- and low-quality habitats. Strictly protected areas have stronger protection mechanisms against destructive anthropogenic activities within their boundaries [71,72] and serve as cushions against deleterious anthropogenic activities around their borders. As national parks and game reserves exclude most forms of human activities [27], habitat quality and intactness in such protected areas are expected to be high. Yet, our findings and those of previous studies suggest that human activities are adversely impacting wildlife habitats in the ecosystem [8,73], reducing the effective size of the protected areas [74], wildlife habitats, and their ecological productivity [74,75]. These adverse effects are more evident in the Ngorongoro conservation area that allows settlement of the Maasai with their livestock (agriculture is currently banned) and in the Loliondo game controlled area that allows settlement, cultivation, and livestock keeping [5,27]. Generally, our findings concur with those of [8] who also documented rapid and unsustainable habitat loss in the Greater Serengeti ecosystem.

The increase in the coverage of poor-quality habitats during 1975–2015 was concurrent with human population growth [5,8,76]. Human population growth, coupled with expanding agriculture, settlements, and livestock, by exerting pressure on natural resources around the ecosystem, are the major drivers of habitat degradation and loss [77]. In particular, the high proportion of poor-quality habitats in the ecosystem by 2015 was likely due to the fact that areas around the Serengeti ecosystem have the highest population growth rate in Tanzania [5]. Human population growth and its associated impacts are also increasingly reducing habitat quality in terrestrial ecosystems elsewhere inside and outside Tanzania [52,57,78].

Improvements in habitat quality in WMAs in the ecosystem are partly attributed to strengthened collaborative and participatory conservation approaches spearheaded by the national and local governments, non-governmental organizations, and local communities. The high habitat quality in the Serengeti National Park and Ngorongoro Conservation Area is associated with the high level

of protection against human-induced threats, implying the importance to biodiversity conservation of effectively protecting habitats against human pressures impinging against their borders [79]. Other studies have similarly inferred the importance of the effectiveness of protection to habitat quality and biodiversity conservation in terrestrial ecosystems worldwide [80–83].

## 5. Conclusions

We mapped and evaluated spatial and temporal changes in wildlife habitat quality (1975–2015) in the GSE and its surrounding buffer area using the Integrated Valuation of Environmental Services and Tradeoffs (InVEST) model to provide a quantitative basis for effective biodiversity management and conservation for the Serengeti and other ecosystems experiencing similar anthropogenic impacts. There was an overall increase in habitat quality with increase in conservation status of protected areas. Consequently, most high-quality habitats were found within strict protected area boundaries. High-quality habitats declined slightly during 1975–1995 but increased during 1995–2015. The ecosystem was dominated by poor-quality habitats but a slight overall increase in medium and high-quality habitats was evident in 2015. Poor-quality habitats increased over time, most especially in the buffer area, whereas protected areas, especially the Serengeti National Park, maintained high-quality habitats throughout 1975–2015. Over the course of the 41 years (1975–2015), the poor-quality habitats increased up to eight-fold in the GSE and the surrounding buffer area.

Anthropogenic activities such as livestock keeping, overgrazing, settlements and agriculture in human- dominated areas, unless checked, will continue adversely impacting the habitat quality and disrupting natural ecological processes. However, strengthening legal and institutional arrangements, such as establishing new and/or upgrading the conservation status of protected areas, would help improve habitat quality. Further research is needed to explore how to effectively enforce the Wildlife Conservation Act (WCA) and WMA regulations and integrate them with other related pieces of legislation to safeguard wildlife habitats in protected areas and their buffer zones. Conservation strategies should also be strengthened in all the protected areas and their surrounding buffer areas to maintain high-quality habitats and healthy wildlife populations.

**Author Contributions:** H.K.K., E.F.N., M.Y.S., and J.R.K. conceptualized and designed the research; H.K.K. collected the data; H.K.K., E.F.N., M.Y.S., and J.O.O. and analyzed the data, and drafted the manuscript; M.Y.S., E.F.N., and J.R.K. supervised the research work; E.F.N., M.Y.S., J.O.O., J.R.K., L.J.M., J.B., B.J.G., and F.V. reviewed and edited the manuscript for critical intellectual content. All authors have read and agreed to the published version of the manuscript.

**Funding:** This research work is a part of the Ph.D. study of H.K.K. and was funded by the European Union's Horizon 2020 research and innovation grant number 641918 through the AFRICANBIOSERVICES PROJECT and by the FRIEDKIN CONSERVATION FUND (FCF). J.O.O. was also supported by a grant from the German National Research Foundation (DFG; Grant # 257734638).

**Acknowledgments:** We thank the Tanzania Wildlife Research Institute (TAWIRI) for granting study leave and permission to H.K.K. to undertake research work, and Tanzania National Park (TANAPA), Ngorongoro Conservation Area (NCAA), Tanzania Wildlife Management Authority (TAWA), and Wildlife Management Area Authorities (WMAs) for granting access to the areas under their jurisdiction for data collection. We also thank Itaely Nassary, Noel Alfred and Mawazo Nzunda for safe driving during data collection and Devolent Mtui for assisting with data collection.

**Conflicts of Interest:** The authors declare no conflict of interest.

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
