# Peer review of "Spatio-Temporal Changes in Wildlife Habitat Quality in the Greater Serengeti Ecosystem"

_sustainability, doi:10.3390/su12062440_

Round 1

Reviewer 1 Report

This is a well written paper and should be of interest to anyone in management and conservation of protected ecosystems. It is my view that the paper is worthy of publication. However, although the approach to analysis seems appropriate to me, I am not familiar with this type of analysis but I assume that other reviewers with more appropriate statistical skills will comment on the analyses if there are any issues. It is for this reason that I have not given my opinion on the analyses or also the overall merit of the paper. 

During my review, I picked up a few very minor corrections that are needed but they are no more then would be expected to be picked up during the production phase. They are as follows:

p46: delete 'a' before 'landscapes'

p53: delete space before 'habitat'

p66: 'expansion' not 'expansions'

p160: delete space after '2015'

p165: delete space around hyphen between 'October' and 'November' to be consistent with punctuation associated with other months mentioned at this point of the paper

p170: I don't think it is valid to reference a paper that has been 'submitted' (i.e., not at least accepted for publication) when the information is required as part of the Methods for this paper

p 212: Insert the name the author/s before the reference at the end of the sentence

p 429 delete space before [8]

p 434 delete space before 'habitat' and delete comma after 'loss'

452 & 453 delete space after the 'stop' at the end of the sentence 

Author Response

I have attached Review 1 comments as I responded

Reviewer 2 Report

This research article used the Integrated Valuation of Environmental Services and Tradeoffs (InVEST) model to evaluate the habitat quality of the Greater Serengeti Ecosystem in northern Tanzania. The study used GIS to evaluate satellite imagery of protected and unprotected areas during three points in time; 1975, 1995, and 2015. Images were classified according to habitat type, threats, legal accessibility, and habitat sensitivity.

This analysis follows the change in habitat over a large area over a 40-year period and anthropogenic impacts and land-use policy changes. The figures present complex data in a way that is easy to understand. The importance of preserving relatively intact ecosystems and the great migrations cannot be stressed enough. Once the great migrations are disrupted, they cannot be recreated. This information gives the evidence to the policy makers to back decisions for increased protection and the impetus to look for new solutions for the human population dynamics. One of the most important features of this paper is reporting the influence of governmental land-use policy on changes in habitat quality.

Line 47: Delete s in landscape

Line 77: worldwide. Delete remainder of sentence (redundant with later sentence).

Line 91: Delete Even so,

Line 96: a previous study

Line 98: change is to was

Line 102: in habitat quality at three points in time (1975, 1995, and 2015)

Line 107: The GSE (spelled out first use only, change throughout manuscript, line 374, 377 etc.). Check to see/decide if it should be capitalized including in the references. This is inconsistent and should be the same throughout the manuscript.

Line 304: capitalize Area

Line 325: Conservation Area plural? National Park plural? This should be consistent if there are more than one.

Figure line 354: would it be possible to shade the grouping labels in groups of three? This would make it more evident that the comparisons are over three time periods without increasing the complexity or size of the figure.

Author Response

Attached is the response to the Reviewer 1
